# OpenReview forum: "Learned Mixing Weights for Transferable Tabular Data Augmentation"
_ICLR.cc/2024/Conference — ICLR 2024 Conference Withdrawn Submission_

### Official Review · Reviewer_fbLq · 2023-10-27

**Soundness:** 2 fair
**Presentation:** 2 fair
**Contribution:** 2 fair
**Rating:** 3
**Confidence:** 3

**Summary:**

The authors propose a method to perform mixup on tabular data. Instead of mixing the features themselves, some features are chosen directly from one sample, some from the other sample. An MLP is used to determine the weighting of the two samples in the loss.

**Strengths:**

The method is simple and widely applicable across architectures. It appears to provide consistent benefits on the majority of datasets used.

**Weaknesses:**

- The paper provides little motivation for many of the aspects of the approach. The writing is not particularly clear, the clarity of the exposition can be much improved still.
- While there are many classification models the method is applied on in the experiments, there is only single data augmentation baseline that is compared against. The difference in performance does not seem significant. The paper also does not contain any comparisons to numbers in the literature, which makes it difficult to judge how good the results are in the first place. Since no per-dataset numbers are provided, it is unclear how large the variance between multiple repeats is. I would especially like to see a comparison against Chen et al. (2023), since they propose a very similar idea where features are swapped between samples (their Feat-Mix variant). The authors would benefit from elaborating on the similarities and differences further than the small paragraph in the current version of the paper, considering the similarity in terms of approach. Chen et al don't evaluate their Feat-Mix with other architectures, but in principle I believe it should apply just the same as the proposed approach here.
- It would be good to have a statistical analysis of whether the differences between models are statistically significant, for example using the Friedman and Nemenyi tests, see [1]

[1] https://www.jmlr.org/papers/volume7/demsar06a/demsar06a.pdf

**Questions:**

Can you specify a and b more precisely in equation 1?
Are the per-sample weights predicted by the MLP consistent between different training runs of the MLP?

---

### Official Review · Reviewer_mN7Y · 2023-10-31

**Soundness:** 3 good
**Presentation:** 3 good
**Contribution:** 2 fair
**Rating:** 5
**Confidence:** 4

**Summary:**

This paper proposes a structure-agnostic data mixing scheme tailored towards unstructured tabular datasets. Two data points are mixed randomly at a 1:7 ratio of features. A feature selection model $g$ is used to assign per-feature weights for each example. The weight of an example for the mixed up vector is the sum of weight of the features used from that example in the mix. These weights are used to construct a loss function during training, with added regularization terms to avoid uniform feature weight assignment. The weights are also employed during evaluation by linearly mixing the prediction with the output from a weighted version of the feature. The work proposes a transfer learning scheme where the feature importance model $g$ is first pre-trained alongside a differentiable MLP (as the main predictor). The importance model is then used on black-box classifiers such as gradient boosted decision trees. The empirical results show that the proposed method applied on top of current SOTA improves the results on standard benchmark datasets.

**Strengths:**

Originality and significance: The use of data mixing with a learned feature importance model for unstructured data (to the best of my knowledge) is novel to this work. The results show improvements as the method is used with different choices of the predictor model (except for logistic regression). A limited study shows that for continuous domains, the proposed method matches the performance of SnapMix (a feature mixing method that uses CAM scores from the main classifier to calculate mixing weights).

Quality and clarity: The paper is mostly easy to follow and the building blocks of the methods are explained clearly.

**Weaknesses:**

Originality and significance: It can be hard to assess the significance of the work since there is no comparison with other baseline methods suited for data mixing with unstructured data. I would have liked to see an ablation study where $g$ is a constant (uniform feature weights, mixing based on the count of features used from each example) and instead the ratio of the mix is varied.

Quality and clarity: The paper has an unusual flow. The introduction section includes many details of the experiments, while the motivations, intuitions and discussions are scattered across other sections. I had to go back and forth several times to find relevant details in various sections. IMO a restructuring of the text can help with readability.

**Questions:**

Notes and questions:

- Mixing at evaluation time is not well motivated in the paper. Please elaborate why you would expect to see a performance boost by mixing with a weighted example prediction.
- If we average $g(.)$ across a dataset, would it result in average feature importance weights that could be used for feature selection, or are these weights mainly useful for example mixing?
- Figure 1 is missing a legend. You can also rearrange and resize to a much smaller footprint without losing clarity.
- SnapMix needs to be explained in more detail, maybe with a short description of CAM.

---

### Official Review · Reviewer_tQjd · 2023-11-01

**Soundness:** 2 fair
**Presentation:** 1 poor
**Contribution:** 2 fair
**Rating:** 3
**Confidence:** 3

**Summary:**

This paper proposes a learnable mixup-style data augmentation technique for tabular classification problems. At a high level, the proposed method learns a weighting function to appropriately weight the contribution of different features in *virtual samples* generated by stitching together different features of different input samples. Empirically, this method improves the performance of many popular learning algorithms.

**Strengths:**

1. The concept of learning a mixing function for mixup-style data augmentation is interesting and novel.
1. Empirically, the proposed method improves classification accuracy for many common algorithms.

**Weaknesses:**

1. The paper opens up with high-level overview of the proposed method and is missing a lot of context. More concretely,  it's missing (1) a clear definition the problem of interest, (2) why the problem is of interest to the community, (3) what methods have prior works proposed and why these methods don't address the problem of interest, and (4) How the proposed method improves upon prior these prior works. Here is my current understanding of the context:
* The problem of interest: training neural models with unstructured tabular data.
* Why the problem is interesting (motivation): unclear.
* Prior works: discussed in related work, but should also be discussed in the intro.
* Core novelty: this is the first work to *learn* a mixing augmentation function for *tabular* problems.
Please let me know if these short descriptions are generally correct. Further detail on these four points would help me better understand the better and also provide more feedback during the review period.

1. I found the method description to be a bit haphazard and hard to follow. A figure illustrating the flow of information in training $g$ and $f$ as well as the method's novelties would be very helpful. The algorithmic details are clear, but the motivation behind these details is unclear to me. Please see the Question section my question related to the algorithm.
1. Section 3 also introduce many new variables which makes it difficult to follow. If it's possible to consolidate notation, that would help immensely with readability. For instance, in Eq 1, the authors may consider replacing $m_i^u$ with $g(x_i)$ and then stating that gradients do not backpropagate through $g$ (analogous for $m_j^u$).
1. It seems the empirical analysis does not compare the proposed mixup method with existing mixup-style methods. This method should at least be compared to a naive, unweighted mixup strategy.

Overall, I vote to reject. The proposed mixup method does seem to accomplish what the paper claims it should accomplish (as seen by Figure 1 and 3), but the paper is quite difficult to follow, much of the algorithm is missing justification, and lacks relevant baselines.

**Other minor comments:**

1. $a$ and $b$ are not defined in Eq. 1.
1. In Table 1, its difficult to identify the top performing methods. I suggest bolding/highlighting the best methods.
1. I prefer the term "augmented sample" over "virtual sample" as is emphasizes that the data was generated using a data augmentation function.
1. Notation for the Hadamard product is defined After Eq 2 in Section 3 but is used throughout the intro.

**Questions:**

1. Could the authors explain the decision to use 7/8 of the features from the first sample? Ablations on this value would shed some light on this decision.
1. I don't quite understand Eq. 3. Why this loss? At a high-level, what is the goal of the learned mixing model? What sorts of mixtures does it aim to generate? Is this loss applied only to virtual samples (if so, please use $\bar{x}$ instead of $x$ here).
1. Is the "weighted sample $dg(x) \circ x$" the sample thing as a virtual sample $\bar{x}$? If so, I suggest rewriting as "virtual sample $\bar{x} = dg(x) \circ x$
1. Why do we want to prevent uniformity?
1. The loss in Eq. 4 can be minimized by updated $g$ such that just a single entry of $m$ large and the rest are uniform. Is this the intended effect?
1. The first contribution mentions that the proposed method operates in the input domain. Could the authors explain why this detail is important?
1. The paper mentions that the proposed method is suitable for unstructured data, but it is not clear to me what is meant by "unstructured." Could the authors please explain?

---

### Official Review · Reviewer_94ej · 2023-11-04

**Soundness:** 2 fair
**Presentation:** 3 good
**Contribution:** 3 good
**Rating:** 6
**Confidence:** 3

**Summary:**

This paper presents a novel data augmentation strategy for tabular data classification. It utilizes an auxiliary network g to learn feature importance for each individual input example. Given a pair of examples, data mixing is realized by replacing 1/8 of one example’s features with the other example’s features. The coefficient of the loss objectives regarding the two examples is determined by the total importance of the mixed features. With the techniques of surrogate networks or gradient-free optimization, the proposed method can be applicable to any classification function. Authors conduct experiments on the AMLB benchmark. Results show that the proposed data mixing strategy outperforms baselines including CatBoost, XGBoost, MLP, TabTransformer, K-NN and Logistic Regression.

**Strengths:**

1. Mixup is a very popular data augmentation scheme. This paper conducts an interesting exploration to apply mixup-like augmentation approaches to tabular data.
2. The proposed idea sounds reasonable. It is model-agnostic.
3. Clear improvements are achieved on most datasets.

**Weaknesses:**

1. As a core component of this paper, feature importance calculation achieved by g sounds a common idea in feature selection. Even if this paper is the first to apply feature selection techniques on tabular data augmentation, the relevant studies should be mentioned.
2. In the experiments, no existing tabular data augmentation methods are evaluated and compared with.
3. It is not clear why feature importance is necessary in the whole method. How about using uniform importance for all features?
4. The improvement on Logistic regression is marginal.

**Questions:**

See weaknesses.

---

### Official Review · Reviewer_6Hg6 · 2023-11-08

**Soundness:** 3 good
**Presentation:** 2 fair
**Contribution:** 2 fair
**Rating:** 5
**Confidence:** 3

**Summary:**

This work proposed an architecture-agnostic method for tabular data augmentation, which features close cooperation between a weight-assigning network and a classifier in both the training and inference phases. Experiments demonstrate the effectiveness of the proposed feature mixing technique on tabular data.

**Strengths:**

- the idea of co-train a weight assignment network $g$ together with a classifier $f$ is interesting. Such a paradigm of close cooperation between two networks may be used in other tasks.
- the proposed technique is effective in improving multiple tabular data classifiers.

**Weaknesses:**

- This paper violates the double-blind policy. The first paragraph of Sec.1 reads, "... we develop a data mixing (Guo et al., 2019; Huang et al., 2021) technique ...".
- The proposed technique is not subject to theoretical analysis, nor is an intuitive picture explained.
    - it is hard to judge the applicability of the proposed technique, learning a network $g$ for data mixing, in general scenarios. It is possible that $g$ overfits and degrades the resulting $f$.
    - the authors claim their technique is "architecture-agnostic." However, for more complex $f$, simple MLP-based $g$ may not be able to capture its behavior (recall that $g$ is trained to replicate the logits of $f$). It is interesting to ask how the architecture of $g$ will affect the performance.
    - why should we expect that the feature weight is beneficial during inference?
- The presentation of ideas could be better organized and clearer.
    - there lacks a description of how to co-train the two networks, $f$ and $g$,
    - In Sec. 3, it is unclear what the authors mean by stating, "... we sum the weights assigned by network g that are associated with the relevant coordinates of each sample". A mathematical expression would be helpful.
        - as features chosen from the two samples are non-overlapping, do the authors mean that we add the corresponding weights of two samples regardless of whether the feature is selected for constructing the virtual sample?
    - Eq. 1, a summation ranges from $u=a$ to $b$. However, there is no definition for $a$ and $b$.
    - Eq. 3, the symbol $dg$ could be confusing; it may be better to write it as $d*g$

**Questions:**

- why should we expect that the feature weight is beneficial during inference?
- the experiments mainly consider the tabular data. It is interesting to ask whether the proposed technique also shows competing performance on different types of data, e.g., images, compared to other feature mixing methods.
- how would the performance change when adopting different architectures for $g$
- how are the two networks, $f$ and $g$, co-trained in detail?
- it seems one can also use a differentiable loss for $\mathcal{L}_m$, e.g., the entropy, which encourages the weights to concentrate on one particular element of $m$.  If adopting, say, the entropy as $\mathcal{L}_m$, how would the resulting network $g$ be affected?